# Standard Quality Characteristics and Efficacy of a New Third-Generation Antivenom Developed in Colombia Covering *Micrurus* spp. Venoms

**DOI:** 10.3390/toxins16040183

**Published:** 2024-04-09

**Authors:** Santiago Tabares Vélez, Lina María Preciado, Leidy Johana Vargas Muñoz, Carlos Alberto Madrid Bracamonte, Angelica Zuluaga, Jeisson Gómez Robles, Camila Renjifo-Ibañez, Sebastián Estrada-Gómez

**Affiliations:** 1Grupo de Toxinología y Alternativas Terapéuticas—Serpentario, Facultad de Ciencias Farmacéuticas y Alimentarias, Universidad de Antioquia (UdeA), Medellin 050010, Colombia; santiago.tabares@udea.edu.co (S.T.V.); maria.preciado@udea.edu.co (L.M.P.); jeisson.gomez1@udea.edu.co (J.G.R.); 2Tech Life Saving (TLS), Tech Innovation Group Company, Medellin 050022, Colombia; leidy.vargasmu@campusucc.edu.co (L.J.V.M.); aseguramientotls@udea.edu.co (C.A.M.B.); angelica.zuluaga@udea.edu.co (A.Z.); 3Facultad de Medicina, Universidad Cooperativa de Colombia, Medellin 050012, Colombia; 4Centro de Investigación Tibaitatá, Corporación Colombiana de Investigación Agropecuaria—AGROSAVIA, Bogota 250047, Colombia; mrenjifo@agrosavia.co

**Keywords:** antivenom, quality control, coral snake, preclinical, freeze-dried

## Abstract

In Colombia, *Micrurus* snakebites are classified as severe according to the national clinical care guidelines and must be treated with specific antivenoms. Unfortunately, these types of antivenoms are scarce in certain areas of the country and are currently reported as an unavailable vital medicine. To address this issue, La Universidad de Antioquia, through its spin-off Tech Life Saving, is leading a project to develop third-generation polyvalent freeze-dried antivenom. The goal is to ensure access to this therapy, especially in rural and dispersed areas. This project aims to evaluate the physicochemical and preclinical parameters (standard quality characteristics) of a lab-scale anti-elapid antivenom batch. The antivenom is challenged against the venoms of several *Micrurus* species, including *M. mipartitus*, *M. dumerilii*, *M. ancoralis*, *M. dissoleucus*, *M. lemniscatus*, *M. medemi*, *M. spixii*, *M. surinamensis*, and *M. isozonus*, following the standard quality characteristics set by the World Health Organization (WHO). The antivenom demonstrates an appearance consistent with standards, 100% solubility within 4 min and 25 s, an extractable volume of 10.39 mL, a pH of 6.04, an albumin concentration of 0.377 mg/mL (equivalent to 1.22% of total protein), and a protein concentration of 30.97 mg/mL. Importantly, it maintains full integrity of its F(ab′)_2_ fragments and exhibits purity over 98.5%. Furthermore, in mice toxicity evaluations, doses up to 15 mg/mouse show no toxic effects. The antivenom also demonstrates a significant recognition pattern against *Micrurus* venoms rich in phospholipase A_2_ (PLA_2_) content, as observed in *M. dumerilii*, *M. dissoleucus*, and *M. isozonus*. The effective dose 50 (ED_50_) indicates that a single vial (10 mL) can neutralize 2.33 mg of *M. mipartitus* venom and 3.99 mg of *M. dumerilii* venom. This new anti-elapid third-generation polyvalent and freeze-dried antivenom meets the physicochemical parameters set by the WHO and the regulators in Colombia. It demonstrates significant efficacy in neutralizing the venom of the most epidemiologically important *Micrurus* species in Colombia. Additionally, it recognizes seven other species of *Micrurus* venom with a higher affinity for venoms exhibiting PLA_2_ toxins. Fulfilling these parameters represents the first step toward proposing a new pharmacological alternative for treating snakebites in Colombia, particularly in dispersed rural areas, given that this antivenom is formulated as a freeze-dried product.

## 1. Introduction

Colombia, due to its geographic characteristics and location, holds a high biodiversity within the suborder Serpentes, with over 334 species inhabiting the country. Among these, 54 species hold medical importance. These snake species are distributed across two families: Viperidae and Elapidae [1,2]. Notably, the most representative genera in both families include *Bothrops*, *Lachesis*, *Crotalus*, and *Micrurus*. In 2022, these genera collectively accounted for 5573 snakebites in Colombia [3]. The *Micrurus* genus (comprising coral snakes from the Elapidae family) was responsible for 1.3% of the total snakebites in the country during that year. Interestingly, the venom from these coral snakes exhibited higher lethality (3%) and severity (7.3%) compared to the average snakebites in Colombia [3].

The *Micrurus* venom comprises two main protein families: three-finger toxins (3FTxs) and PLA_2_, exhibiting a dichotomy across the Americas [4]. In Colombia, 3FTxs are dominant in certain species, such as *M. mipartitus* [5], while PLA_2_s prevail in other species, like *M. dumerilii* [6]. These two coral snake species hold the most significant epidemiological importance in the country and are primarily distributed in the Andean zone [7].

The envenoming symptomatology associated with coral snakebites is highly conserved in almost all *Micrurus* envenomation and consists of flaccid paralysis that can progress and cause respiratory failure [8,9,10]. These manifestations are set after the postsynaptic interactions of 3FTxs with acetylcholine receptors, as well as the presynaptic activity of neurotoxic PLA_2_s [11]. Since both toxins can be distributed (albeit in different concentrations) in most *Micrurus* individuals in Colombia, it becomes challenging to identify the aggressor species based solely on symptomatology. This underscores the necessity of producing polyvalent antivenoms in the country.

Antivenoms represent the only scientifically proven pharmacological therapy for treating snakebites, and their access should be mandatory, particularly in rural dispersed areas where over 70% of snakebites occur annually in Colombia [12]. Unfortunately, the production of anti-elapid antivenoms in the country could be insufficient and irregular due to limited production, challenges in accessing venoms (owing to the fossorial and slippery behavior of *Micrurus* species), and low survival rates during the captivity of *Micrurus* individuals [13,14]. Consequently, antivenom shortages have been reported at different times in Colombia, and currently, anti-elapid antivenoms are classified as vital medicine unavailable due to limited access. Furthermore, the current available anti-elapid antivenom in the country is in liquid form, necessitating a cold chain for distribution and storage, which restricts access in rural dispersed areas [15]. This situation led to the inadequate treatment of snakebites in 2022, where 45.8% of coral snakebite patients did not receive antivenoms [3], and the treatment that these patients received was not specified in the reports.

The currently available anti-elapid antivenom in Colombia is a second-generation product produced by the National Health Institute (INS). These second-generation antivenoms are composed of complete immunoglobulins G (IgG) obtained through an immunization process in horses. In contrast, third-generation antivenoms consist of F(ab′)_2_ fragments obtained from the IgG using pepsin. The removal of the IgG Fc fraction reduces its size [16], and some authors have proposed that these antivenoms exhibit fewer adverse reactions and improved pharmacokinetics [17,18].

To enhance and ensure access to this treatment, the Universidad de Antioquia, through its spin-off Tech Life Saving, has developed a third-generation freeze-dried antivenom (specifically targeting elapid snakes). This formulation aims to meet the quality control parameters outlined in the WHO guidelines for antivenom production [19].

The quality control analyses involve assessing standard quality characteristics (including physicochemical characterization) and conducting preclinical tests using a murine model with both the antivenom and the venoms used in the test process. Consequently, this study seeks to implement these quality control analyses for the lab-scale product, ensuring compliance with international quality standards set by the WHO [19].

## 2. Results

### 2.1. Venoms Analysis

#### 2.1.1. Diversity in Venom Composition

To determine the most probable composition of the venoms, electrophoresis and reverse-phase high-performance liquid chromatography (RP-HPLC) was used. This method revealed a complex profile in all venoms, with most compounds eluting between 15% and 40% of acetonitrile (ACN). Specifically, we collected 16 peaks from *M. mipartitus* venom (Figure 1A), 36 peaks from *M. dumerilii* venom (Figure 1B), 27 peaks from *M. spixii* (Figure 1C), *M. dissoleucus* (Figure 1E), *M. isozonus* (Figure 1F), *M. medemi* (Figure 1G), and *M. lemniscatus* (Figure 1H) venoms, 36 peaks from *M. ancoralis* (Figure 1I) venom, and 23 peaks from *M. surinamensis* (Figure 1D). Subsequently, all the peaks underwent sodium dodecyl sulfate–polyacrylamide gel electrophoresis (SDS-PAGE) analysis. Notably, components of the protein families 3FTXs and PLA_2_s were identified in all venoms.

#### 2.1.2. Venoms Lethality

To assay the biological activity of the venoms and be able to analyze the neutralization capacity of the antivenom, the lethality of *M. mipartitus* and *M. dumerilii* venoms were confirmed using three times the lethal doses 50 (LD_50_) published by Otero et al., 1992 [22] and Rey-Suárez et al., 2016 [6], respectively. The results from both 3xLD_50_ showed 100% lethality in mice (weight 18–20 g) within the first 24 h. Specifically, the confirmed dose for *M. mipartitus* was 27 µg venom/mouse (equivalent to 1.35 mg/kg), and for *M. dumerilii* venom, it was 114 µg venom/mouse (equivalent to 5.7 mg/kg). Using this information, we proceeded with efficacy tests on the two species of main epidemiological importance in the country.

### 2.2. Antivenom Physicochemical Characterization

#### 2.2.1. Antivenom Appearance, Solubility Time, Extractable Volume, and pH

To determine some standard quality characteristics of the antivenom, six antivenom vials (batch number: AVM-P191220022) were resuspended and analyzed to determine their appearance, solubility time, extractable volume, and pH (Figure 2 and Table 1). The freeze-dried antivenom exhibited a white, uniform, and compact powder-like appearance, without any visible particles of any color in the vial. After resuspension, the product became completely translucent, with no suspended visible particles or turbidity (Figure 2). Upon adding 10 mL of Water for Injection (WFI) and gently shaking, the antivenom fully reconstituted in 4 min and 25 s ± 0.17 s, and the extractable volume was measured as 10.39 mL ± 0.19 mL. The average pH was 6.04 ± 0.05 (Table 1).

#### 2.2.2. Antivenom Total Protein and Albumin Concentration

To determine some chemical characteristics of the antivenom, the six previously resuspended vials were tested using the Beirut methodology to determine the total protein content. The results revealed an average protein concentration of 30.97 mg/mL (Table 2). In comparison, the second-generation liquid antivenom from the INS of Colombia exhibited a total protein concentration of 42.68 mg/mL.

In the six vials, the average concentration of albumin was tested and found to be 0.38 mg/mL, accounting for 1.225% of the total protein in the product. In contrast, the albumin concentration for INS was 1.05 mg/mL, representing 2.46% of the total protein in their product (Table 2). Notably, the albumin concentration of our product was lower when compared to the commercial product available in the country.

#### 2.2.3. Antivenom Purity, Integrity, and Molecular Size Distribution

To determine the molecular size, purity, and integrity of the components in the antivenom, a size-exclusion chromatography (SEC-HPLC) profile was performed. In this profile, two peaks were detected: the first one with higher intensity, an elution time of 19.08 min, and a purity index of 99.7%, and the second one with an elution time of 24.75 min and a purity index of 0.3%.

Similarly, the SDS-PAGE electrophoretic profile revealed a predominant band with a calculated molecular weight (MW) of 115 kD. Additionally, at least four more bands were detectable, each with a calculated MW very close to that of albumin in one case. In contrast, the second-generation commercial antivenom from the INS, used as a reference, exhibited a predominant band with an MW of 175 kD (Figure 3), along with four other smaller bands. These findings indicate the high purity and integrity of the F(ab′)_2_ component in the product.

#### 2.2.4. Antivenom Identity (Ouchterlony)

To determine qualitative recognition, the antivenom was checked against the venoms of the nine *Micrurus* species in a ratio of 1 ug of venom to 30 ug of antivenom. The antivenom exhibited a strong recognition for the venoms from *M. mipartitus*, *M. dumerilii*, *M. isozonus*, *M. dissoleucus*, and *M. ancoralis*. However, the venom recognition for *M. ancoralis* and *M. dissoleucus* was less intense and less detectable, with fainter recognition stripes. Notably, the test did not show recognition for the venoms of *M. lemniscatus*, *M. spixii*, *M. medemi*, and *M. surinamensis* at the evaluated concentrations (Figure 4). This suggests better recognition in general for the venom components dominated by PLA_2_s proteins.

### 2.3. Preclinical Tests

#### 2.3.1. Antivenom Binding Titer

To determine the quantitative value for the recognition of the antivenoms against the venoms, an enzyme-linked immunosorbent assay (ELISA) was used. Compared to the other venoms, the highest antivenom recognition titers were obtained from *M. isozonus* 1:10,000 (with absorbances 0.078 ± 0.009 and cut-off 0.069 ± 0.016), *M. dissoleucus* 1:12,000 (with absorbances 0.085 ± 0.01 and cut-off 0.058 ± 0.008), and *M. dumerilii* 1:12,000 (with absorbances 0.043 ± 0.004 and cut-off 0.036 ± 0.023). Some venoms exhibited mild recognition: *M. spixii* 1:4000 (with absorbances 0.075 ± 0.003 and cut-off 0.069 ± 0.017), *M. medemi* 1:2000 (with absorbances 0.42 ± 0.02 and cut-off 0.31 ± 0.05), *M. lemniscatus* 1:2000 (with absorbances 0.064 ± 0.009 and cut-off 0.067 ± 0.016) and *M. ancoralis* 1:2000 (with absorbances 0.098 ± 0.005 and cut-off 0.068 ± 0.009). In contrast, the venoms from *M. surinamensis* 1:50 (with absorbances 0.08 ± 0.01 and cut-off 0.052 ± 0.006) and *M. mipartitus* 1:1000 (with absorbances 0.16 ± 0.01 and cut-off 0.13 ± 0.017) showed lower recognition capacity by the antivenom (Figure 5). Overall, the titer was higher for venoms primarily composed of PLA_2_s proteins.

#### 2.3.2. ED_50_

To determine the efficacy against the venoms of the two species of primary epidemiological importance in the country, an additional freeze-dried vial of the antivenom was used to determine the ED_50_ for the final formulation through testing in mice. A volume of 1 mL of the polyvalent freeze-dried formulation was challenged against 3xLD_50_ of *M. mipartitus* and *M. dumerilii*, and neutralized 232 µg and 399 µg of venom, respectively (Table 3). This means that 10 mL of the freeze-dried antivenom vial can neutralize 2.32 mg of *M. mipartitus* and 3.99 mg of *M. dumerilii* venoms. Notably, all positive controls died within 48 h, while the negative controls survived. Considering an effective dose 100 (ED_100_), the full potency of the freeze-dried antivenom vial can neutralize 1.55 mg of *M. mipartitus* and 2.66 mg of *M. dumerilii* venoms.

#### 2.3.3. Acute Toxicity

The safety of the product was tested in mice, and no evidence of toxicity was found at any of the three doses tested (9 mg, 12 mg, and 15 mg of antivenom). We observed no changes in any of the parameters evaluated during the toxicity test, and there were no visible alterations in any of the analyzed organs (the observed parameters are listed in Appendix A). Additionally, all *p*-values in the comparative t-test were greater than 0.05% when comparing the organ weights of the animals to the reference values [24] (see Table 4). Consequently, we can conclude that the product itself does not induce adverse reactions in mice.

## 3. Discussion

*Micrurus* venom is currently a crucial and limited resource for producing and assessing the efficacy of antivenoms (i.e., determining how much venom the product would be able to neutralize). The basic methodology to obtain the venom is to collect live animals in the wild and keep them in captivity, continuously extracting venom manually until their death. However, this process is particularly difficult for *Micrurus* genus snakes due to their fossorial and nervous behavior [8].

Of the nine species evaluated from Colombia in this study, only four proteomic reports are available (*M. mipartitus*, *M. medemi*, *M. lemniscatus*, and *M. dumerilii*). These reports indicate that across the American continent, *Micrurus* venoms express two main proteins involved in the clinical manifestations after snakebite: 3FTx and PLA_2_ [21]. According to Rey-Suárez et al. (2011) [25], *M. mipartitus* is an elapid that contains a significant amount of 3FTxs. It expresses a neurotoxin called Mipartoxin, which serves as the major component in its venom. Mipartoxin is easily identifiable in the chromatographic profile due to the intensity of the peak and its low hydrophobicity [25]. Our chromatographic and electrophoretic results for *M. mipartitus* are consistent with the previous report of Rey-Suárez et al. (2011) [5,25]. We were able to detect the main fraction corresponding to the Mipartoxin (peak number 4 on the RP-HPLC), and both the ACN elution percentage and the MW showed remarkable similarity to the report by Rey-Suárez et al. [6]. Conversely, *M. dumerilii*, *M. lemniscatus*, and *M. medemi* show a venom content dominated by PLA_2,_ as previously described by Rey-Suárez et al. (2016) [6], Sanz et al. (2019) [21], and Rodríguez-Vargas et al. (2023) [26], respectively.

The only available venomic reports for *M. spixii* and *M. surinamensis* come from Brazil, where Sanz et al., 2019 [20] identified that *M. spixii* venom expresses a majority composition of 3FTxs. Although this composition is relatively in balance with PLA_2_s, 3FTxs is dominant in *M. surinamensis* [20]. Our findings from chromatographic and electrophoretic profiles of *M. spixii* and *M. surinamensis* showed a highly similar chromatographic profile with a MW distribution of their proteins in a very close rank, allowing us to propose a possible distribution of protein families within each venom. However, in the literature reviewed, no proteomic reports were found for *M. ancoralis*, *M. dissoleucus*, and *M. isozonus.* Consequently, we were unable to propose a possible distribution of protein families for these venoms. Nevertheless, based on the HPLC profile and electrophoresis pattern, we have a rough idea about the composition of these venoms. According to our observations, all four venoms are likely PLA_2_s-predominant. Nonetheless, further proteomic analysis is necessary to confirm these findings.

Snake antivenoms are the sole pharmacological therapy accepted by the WHO for the treatment of snakebite envenoming. Their safety primarily relies on the quality of the product, which is confirmed through rigorous quality control analyses. The WHO has published guidelines aimed at harmonizing and ensuring antivenom quality and safety (specifically, Annex 5 of WHO Technical Report Series, No. 1004, 2017). In Colombia, local antivenom manufacturing policies are aligned with these guidelines (Decree 821 of 2017). The manufactured antivenom produced by the Universidad de Antioquia and its spin-off Tech Life Saving (batch: AVM-P191220022) conforms to the WHO’s guidelines. It exhibits no particles or different substances in the product, indicating a robust purification and filtration process. Furthermore, solubilization of a single vial occurs within an average of 4 min and 25 s, consistent with the WHO’s recommended maximum limit of 10 min. This result evidenced an accurate process of freeze-drying, which is critical for ensuring product availability in regions where maintaining a cold chain for liquid products is difficult. The recovery volume (extractable volume) after reconstitution exceeds 10 mL, signifying a proper solubilization process.

In line with WHO guidelines, the protein concentration also meets the maximum limit of 100 mg/mL, with a concentration of 30.97 mg/mL. This parameter is particularly crucial because it directly impacts the product’s potency and the likelihood of adverse reactions following administration [27]. Antivenoms must strike a balance in protein concentration to achieve the highest potency while minimizing the probability of adverse reactions. This involves maximizing the quantity of the active pharmaceutical ingredient (API) while minimizing other proteins, such as albumin. Such optimization can be achieved if the protein content of the respective API, in this case, F(ab′)2 fragments, is majoritarian. However, some studies have demonstrated significant variability in this parameter, ranging from 32 mg/mL [28] to 60 mg/mL [29].

We found two parameters deviated regarding WHO recommendations: pH and albumin concentration. According to the WHO, the pH should ideally be close to 7.0; however, the pH for the antivenom product measured 6.04. Despite this deviation, the obtained value may not pose an issue with antivenom administration. Solano et al. (2012) [30] stated that the low pH does not significantly impact the safety and efficacy of the antivenom; on the contrary, a low pH formulation could aid in preserving protein integrity [31]. Nonetheless, because of the high composition stability of a freeze-dried product, it is recommended to adjust the formulation to the pH level suggested by the WHO.

Additionally, future batches of the product should undergo an adjustment in albumin content. While the WHO recommends setting a concentration of albumin under 1% of the total concentration of proteins in the product, the antivenom tested currently presents an average of 1.2% albumin. Furthermore, it is important to consider that this value is 50% lower than the albumin concentration found in the current commercial antivenom available in Colombia (INS liquid antivenom). This parameter is of utmost importance, as albumin is the primary antivenom contaminant and is responsible for adverse reactions after antivenom administration [32]. When the albumin concentration is reduced in the final product, the likelihood of secondary effects decreases, thereby enhancing product safety and increasing community trust, which is crucial for treatment acceptance [33]. In addition to albumin, the electrophoretic profile reveals other impurities present in low concentrations. Among these, some may be residues of pepsin and other byproducts from the purification and separation process of the F(ab′)2 fragments. It is also necessary to reduce these impurities in the subsequent batches of the product.

Protein and albumin concentration are directly related to the antivenom’s purity and integrity of the API. In this case, the antivenom API demonstrates a high level of purity (>98.5%), as confirmed by both SEC-HPLC (99.6%) and albumin determination (98.8%) in relation to the total protein concentration. These findings indicate that the analyzed antivenom contains the necessary proteins to achieve the desired potency with minimal adverse reactions (as evidenced by ED_50_ and toxicity tests). The purity assessment was based on two results, not solely the SEC-HPLC. While the SEC-HPLC provided a close estimate of purity, we observed only two peaks in the chromatogram. In contrast, electrophoresis revealed at least three distinct bands. Additionally, we considered the albumin concentration, which represented the lower end of purity concerning protein concentration.

In addition to the physiochemistry assays, three preclinical parameters were satisfactorily evaluated based on WHO recommendations against nine *Micrurus* venoms: identity, abnormal toxicity, and ED_50_. In identity recognition, using the Ouchterlony technique and ELISA, the antivenom recognized all venoms (including *M. mipartitus*) with a “preference” over PLA_2_s-predominant venoms. The higher recognition capacity for the species *M. dumerilii*, *M. isozonus*, and *M. dissoleucus* (Figure 5A) and the mild recognition capacity for *M. lemniscatus* and *M. medemi* (Figure 5B) may be related to the PLA_2_s-content and the predominance of different PLA_2_s sub-types within each of these venom species. Although *M. ancoralis* may also be a PLA_2_s-predominant venom, the antivenom recognized it strongly at higher concentrations; however, this effect decayed rapidly after dilution with no clear explanation. This species may also have a PLA_2_s-predominant profile, with a high presumed balance with 3FTxs.

The lower recognition observed for the species *M. spixii* and *M. surinamensis* may be directly related to their dominant contents of 3FTxs. However, the antivenom was able to recognize *M. Mipartitus* venom (a 3FTxs-rich venom) (Figure 5C). The recognition of *M. mipartitus* venom is particularly remarkable because of its epidemiological relevance for Colombia [34]. The findings suggest that the antivenom recognized the predominant PLA_2_s venoms better; however, the recognition varied within this protein family. The ability to strongly recognize PLA_2_s-dominant venoms and weakly recognize 3FTxs venoms, while still recognizing *M. mipartitus* venoms, is directly related to how the immunization process was performed and which venoms were used. In any case, it is important to consider immunization using not only PLA_2_s-dominant venoms but also 3FTxs-dominant venoms, specifically *M. mipartitus*, since it is a medically important species in Colombia. What is remarkable in these results is the capacity of the antivenom to recognize different species of *Micrurus* venoms.

The abnormal toxicity evaluation using mice complies with WHO recommendations. The antivenom did not cause any animal variation that can be attributed to a negative reaction to the product. These findings suggest that the impurity concentrations, including albumin, may not cause an adverse reaction in an average application at the evaluated doses. Nonetheless, it is still advised to refine the process of purification for the F(ab′)_2_ in the product. This result is consistent with the purity and integrity analyses, where the obtained product showed a very low concentration of albumin contamination.

The antivenom successfully neutralized the venom from the most important *Micrurus* snake in Colombia. Regarding the commercially available product in Colombia, the antivenom from Universidad de Antioquia showed a lower neutralization capacity, but it was still enough to cover any snakebite by *M. mipartitus* and *M. dumerilii*. Each vial can neutralize up to 2.32 mg and 3.99 mg of each venom, respectively, based on the ED_50_.

This new product is evidence of the capacity of both the public and private sectors to develop efficient biological products. In this case, it is a polyvalent third-generation freeze-dried antivenom. This work demonstrates the antivenom formulation’s ability to cross-react with various *Micrurus* snake species, opening up the possibility of obtaining a polyvalent antivenom product that covers many species within the *Micrurus* genus.

## 4. Conclusions

This new polyvalent freeze-dried third-generation antivenom from Universidad de Antioquia and its spin-off Tech Life Saving meets most of the quality parameters recommended by the WHO, including appearance, solubility time, extractable volume, protein concentration, purity, and integrity. It did not exhibit abnormal toxicity, even at high concentrations. However, the antivenom product should be formulated at a higher pH than the current level, and efforts should be made to reduce the albumin concentration and other potential contaminants.

The antivenom recognizes the venoms of at least nine different *Micrurus* species, including the most important ones in the country’s epidemiology. It exhibits better recognition of the PLA_2_s proteins than the 3FTxs. The product is capable of neutralizing the venom of *M. mipartitus* and *M. dumerilii*. Each vial can protect up to 3.99 mg of *M. dumerilii* venom and 2.32 mg of *M. mipartitus* venom. Considering an ED_100,_ the full potency of the freeze-dried antivenom vial can neutralize 1.55 mg of *M. mipartitus* venom and 2.66 mg of *M. dumerilii* venom.

These results indicate an adequate immunization and production process using specific venoms. This process allows for the production of immunoglobulins capable of recognizing and neutralizing the different venoms of the Micrurus species in Colombia. Additionally, the production process is able to separate, purify, concentrate, and formulate specific F(ab′)_2_ freeze-dried fragments.

## 5. Materials and Methods

### 5.1. Animals

This study used a Swiss strain of mice (*Mus musculus*), both male and female, for in vivo testing. The animals’ weights ranged from 18 to 20 g. All mice were maintained in conditions of 12 h of light and 12 h of darkness. Before and during the experiments, the animals had ad libitum access to food and water. For the experiments that included mice, the three Rs guidelines (Replacement, Reduction, and Refinement) were followed [35]. The protocols were endorsed by the ethics committee for animal experimentation of Universidad de Antioquia, with record number 110.

### 5.2. Antivenom

The antivenom for evaluation was provided by Universidad de Antioquia through its spin-off Tech Life Saving as part of the project to develop a polyvalent freeze-dried third-generation antivenom for Colombia. The provided vials were part of a laboratory-scale batch manufactured in a non-GMP (Good Manufacturing Practice) facility but under a controlled process. The production process consists of the immunization of horses (the information about the venoms used was not supplied), followed by blood extraction, plasma separation, salt precipitation, thermocoagulation, ultrafiltration, sterilization, and lyophilization. Ten vials of the antimicruric antivenom with the batch number AVM-P191220022 were provided.

### 5.3. Venom Obtention

*Micrurus mipartitus* and *Micrurus dumerilii* venoms were obtained in the Serpentarium of the Universidad de Antioquia. The venom was manually extracted (milking) from animals kept under captivity conditions (permission of the collection framework number 524). These venoms were collected and pooled from different animals of the same species and geographic zones (all animals came from the northwest of Colombia). Subsequently, these pools were freeze-dried and stored at a temperature of −20 °C.

Additionally, *M. isozonus*, *M. dissoleucus*, *M. ancoralis*, *M. lemniscatus*, *M. spixii*, *M. medemi*, and *M. surinamensis* venoms were obtained from the venom bank of the antivenom project of Universidad de Antioquia, supplied by María Camila Rengifo.

### 5.4. Venom Analysis

#### 5.4.1. RP-HPLC and SDS-PAGE Analysis

Venoms were analyzed by RP-HPLC using a Shimadzu Prominence chromatograph CBM-20 (pump unit). One mg of crude venom was dissolved in 200 μL of solution A (0.1% trifluoroacetic acid in water) and centrifuged at 3500× *g* for 5 min at room temperature. The supernatant was injected using a C18 RP-HPLC analytical column (250 × 4.6 mm), balanced and eluted at a flow rate of 1.0 mL/min: first isocratically (5% B for 5 min), followed by a linear gradient of 5–15% B for 10 min, 15–45% B for 60 min, and 45–70% B for 12 min. The chromatographic separation was monitored at 215 nm, and fractions above 100 mAU were manually collected (peaks under this level do not present relative abundance enough to represent a relevant venom component) and analyzed using 15% SDS-PAGE according to Laemmli [36] and the gel was stained with FastGene Q-Stain (Cat FG-QS1). Masses of 20 μg of collected peaks protein samples were loaded at a concentrations of 1 µg/µL and a final volume of 20 µL, and a Precision Plus Kaleidoscope (Bio-Rad. Hercules, CA, USA) was used as the standard for estimating MWs with markers covering the mass range from 250 kDa down to 10 kDa. Further, the calculation of MWs was performed using the software GelAnalyzer 19.1, available at http://www.gelanalyzer.com/ (accessed on 2 February 2022) [23]. The software determined the Rf (retention factor, measured as the band distance migrated/gel length) for each analyzed band. It estimated the corresponding MWs using a standard Precision Plus Kaleidoscope (Bio-Rad) with an exponential fit approximation.

Using the MW and the ACN percentage of elution, the fractions were assigned to probable family proteins, following the previous records of the venom composition of each species.

#### 5.4.2. Lethality Confirmation

The lethality of venom was confirmed only for the most epidemiologically important *Micrurus* snakes for Colombia: *M. mipartitus* and *M. dumerilii*. For each confirmation, a group of three mice was used (including a negative control group into which 200 µL of saline solution were injected). Each group was injected intraperitoneally with three times the reported LD_50_ in the literature, in 200 µL of saline solution, and after 24 and 48 h, we evaluated their survival rate. Specifically, for *M. mipartitus*, 27 µg of venom was tested, and *M. dumerilii*, 114 µg of venom was tested, according to Otero et al., 1992 [22] and Rey-Suárez et al., 2016 [6], respectively.

### 5.5. Antivenom Physicochemical Characterization

The standard quality characterization of the antivenom product followed eight of the items recommended by the WHO in Annex 5: Guidelines for the production, control, and regulation of snake antivenom immunoglobulins [19]. The results were then compared with the suggestions presented in those guidelines. The characterization process included assessing appearance, solubility, extractable volume, pH, identity, albumin concentration, protein concentration, immunoglobulin purity, and integrity, as well as molecular size distribution.

#### 5.5.1. Antivenom Appearance, Solubility, Extractable Volume, and pH

The appearance was determined through visual inspection using six vials. Black and white surfaces were employed to contrast the product before and after reconstitution, and the presence of any external particle (different from the white powder) was carefully analyzed and recorded before the product vials were mixed with 10 mL of WFI to reconstitute the freeze-dried content. The mixture was gently shaken until 100% dissolution of the product (at room temperature) and the time was recorded using a standard chronometer. The extractable volume was then analyzed using the resuspended product in the six vials. This volume was collected using a sterile syringe, and the total volume extracted was determined by weighing it on an analytical balance (Precisa EP-225SM-DR. Dietikon, Switzerland). Finally, the antivenom pH was determined using a calibrated pH meter (Orion model Star A221, Thermo Scientific. Waltham, MA, USA) at room temperature (26 °C). This parameter was assessed based on the six reconstituted vials of the product.

#### 5.5.2. Total Protein Concentration

Samples from the previous six vials of antivenom were tested using the Biuret method to determine the protein concentration. To do so, we used the Biosystems Total Protein reagent and Bovine Serum Albumin (BSA) (Biosystems ref 11500) [37,38]. Briefly, 750 µL of Biuret reagent was added to 250 µL of an antivenom sample (after reconstitution using 10 mL of WFI). The mixtures were gently mixed and incubated for 30 min at 37 °C. After incubation, the absorbance was measured at 540 nm. To calculate the protein concentration, we used a calibration curve with different albumin concentrations ranging from 0.1 mg/mL to 50 mg/mL, working under the conditions previously described (y = 0.0798X + 0.0102. R^2^ = 0.9919). All tests were performed in triplicate, and the average absorbances were used for the analysis. Once the sample was read, the value was interpolated in the calibration curve to measure the protein quantity.

#### 5.5.3. Albumin Concentration

To determine the albumin concentration, samples of the previous six vials of antivenom were tested using the method described by Doumas et al. [39] using a specific commercial kit for albumin determination (Biosystems Albumin Reagent, Ref 11547). Briefly, 10 µL of each antivenom (after reconstitution using 10 mL of WFI) were mixed with 1 mL of albumin reagent. The mixtures were gently shaken, and the absorbance was measured at 630 nm. To calculate the albumin concentration, we used a calibration curve with different albumin (BSA) concentrations ranging from 5 mg/mL to 50 mg/mL, working under the conditions previously described (y = 0.0077X + 0.02016. R^2^ = 0.9992). All tests were performed in triplicate, and the average absorbances were used for the analysis. Once the sample was read, the value was interpolated in the calibration curve to measure the albumin quantity.

#### 5.5.4. Identity

The identity was tested using the double immunodiffusion technique described by Ouchterlony [40] and nine *Micrurus* venoms were challenging against the antivenom product. Six different 30 µL wells were drilled in agarose 1% placed on an acetate sheet. In the peripheral wells, 30 µL of reconstituted antivenom (with protein concentrations of 30 mg/mL) were placed, while the central well contained 30 µL of venom (with a concentration of 1 mg/mL). Sheets were incubated for 24 h at 37 °C. After incubation, data were analyzed, and in a positive recognition, the venom and antivenom reacted to form a smooth line of precipitate.

#### 5.5.5. Immunoglobulin Fragments Purity and Integrity

To assess the purity and integrity of the immunoglobulin fragments, we employed 12% SDS-PAGE to analyze the antivenom following Laemmli’s method [41] and stained with the gel with FastGene Q-Stain (Cat FG-QS1). The antivenom samples were loaded at a concentration of 1 µg/µL and a final volume of 20 µL, and a Precision Plus Kaleidoscope (Bio-Rad) was used as a standard for estimating MWs with markers covering the mass range from 250 kDa down to 10 kDa. Further, MWs were calculated using the GelAnalyzer 19.1 software, available at http://www.gelanalyzer.com/ (accessed on 2 February 2022) [38]. The software determined the Rf for each analyzed band through an exponential fit approximation, measuring the distance migrated relative to the gel length.

#### 5.5.6. Molecular-Size Distribution

To determine the purity and integrity of the immunoglobulin’s fragments, we used an SEC-HPLC method using a Sec 3000 Phenomenex column. The mobile phase consisted of buffer Na_2_HPO_4_ (15 mM) NaH_2_PO_4_ (30 mM) NaCl (200 mM). The flow rate was 0.5 mL/min and UV detection was performed at 280 nm.

### 5.6. Preclinical Tests

The preclinical tests were conducted following the WHO guidelines outlined in Annex 5: Guidelines for the production, control, and regulation of snake antivenom immunoglobulins [19]. These tests assess venom binding capacity (titers), ED_50_, and acute toxicity.

#### 5.6.1. Venom Binding Capacity (Titers)

The titers recognition capacity of the antivenoms was determined by ELISA, following the method described by Otero-Patiño et al. [42], with some modifications. In this study, all nine Micrurus venoms were challenged against the antivenom. Plates (Corning Costar microplates ref 3591) were coated overnight, at 4 °C, with 100 µL/well of each venom (0.1 mg/mL in Carbonate/Bicarbonate buffer, pH 9.6). After a washing step, the remaining binding sites were blocked with coated buffer containing 2% BSA (Bovine Serum albumin, Low Heavy Metals US112659-100GM Albumin, Calbiochem) for 1.5 h at 37 °C, followed by a washing step. Then, the antivenom was diluted in sample buffer (PBS pH 7.4, containing 1% BSA.) in concentrations from 1:100 to 1:40,000. Then, 100 µL of each antivenom dilution was added for each well, incubated for 1.5 h at 37 °C, and finally washed. Anti-horse IgG diluted 1:8,000 in sample buffer was added (100 µL/well), incubated for 1.5 h at 37 °C, and washed. Finally, 100 µL O.P.D. -ortho-phenylenediamine (Amresco, Solon, OH, USA) diluted at 0.1% in citrate buffer, pH 6.0, containing 0.1% hydrogen peroxide, was added. Plates were protected from the light and incubated for 20 min at 37 °C, and absorbances were recorded using a Thermo Scientific Multiskan FC microplate (Thermo Scientific. Waltham, MA, USA) reader at 450 nm. Each dilution was tested in quadruplicate using two separate vials (8 repetitions in total); the average of the absorbance was used to determine the titers for each venom. The titers are expressed as the dilution of the antivenom that presents major recognition compared with a cut-off. The cut-off point was set at three times the recognition capacity observed in the plasma of a non-immunized horse.

#### 5.6.2. ED_50_

The ED_50_ was analyzed against the most epidemiologically important snakes. To determine the ED_50_, we used 3xLD_50_ of *M. mipartitus* and *M. dumerilii* venoms, which were incubated for 30 min at 37 °C with different antivenom concentrations. For *M. mipartitus*, 1 mL of antivenom was incubated with 100 µg, 200 µg, and 400 µg of venom. For *M. dumerilii*, 1 mL of antivenom was incubated with 250 µg, 350 µg, 450 µg, and 600 µg. After the incubation, the mixture was injected into groups of three mice. The survival of the animals was recorded at 24 and 48 h. This information was used to calculate the ED_50_ of the products behind different venoms using the statistic methodology Probit [43]. Additionally, a confirmation test was conducted in groups of four mice for each venom. Finally, the full potency based on the ED_100_ was analyzed using the equation ED_100_ = ED_50_/(LD_50_/(LD_50_ − 1)).

#### 5.6.3. Acute Toxicity

The acute toxicity was determined using three groups of three mice, each receiving different doses of antivenom, along with one additional group of three mice serving as the negative control. The antivenom doses were administered intraperitoneally, and the animals were closely monitored for the initial four hours and the subsequent seven days. The parameter register is in Appendix A. A change in a parameter or deaths were registered. The recorded parameters are detailed in Appendix A. Any changes in these parameters or occurrences of mortality were meticulously documented. The negative control group consisted of three mice injected solely with the saline solution used for antivenom dilution at the same volume. After the seven-day period, the animals were dissected, and the weights of their heart, spleen, kidneys, and liver were measured to assess any organ alterations.

For dose calculations, we used weight-adjusted maximum doses relevant to humans based on similar products. Additionally, we adhered to the maximum recommended injection volume for mice. The specific doses administered were as follows: 300 µL, 400 µL, and 500 µL of antivenom, corresponding to 9000 µg, 12,000 µg, and 15,000 µg of antivenom protein.

### 5.7. Statistical Analisis

Physicochemical results are expressed as mean ± standard error of the mean (SEM). These averages and SEM were calculated using the Excel software (Microsoft Office 365). The calibration curves for quantifications and *t*-tests for mean comparison were also created using the same software and confirmed before using the Scipy 1.12.0 package in the Python 3 programming language. The titers were calculated and plotted using Python 3 with the Matplotlib 3.4.3 library for graphics, and the deviation propagation was calculated using the Uncertainties 3.1.7 package. Additionally, in vivo tests were analyzed using the Statsmodels library for probit analysis. Furthermore, the calculation of MW in gels was performed using the GelAnalyzer 19.1 software.

## Figures and Tables

**Figure 1 toxins-16-00183-f001:**
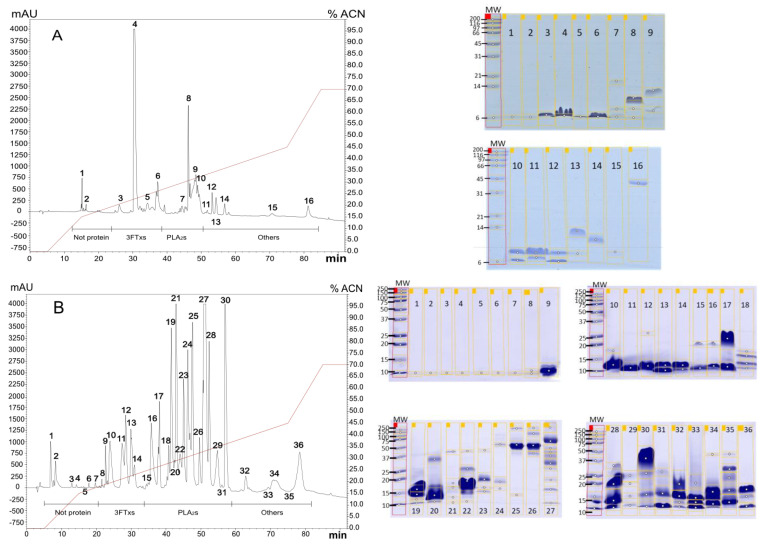
HPLC chromatographic profiles where each peak is numbered and corresponds to a lane in its respective 15% sodium dodecyl sulfate–polyacrylamide gels (SDS-PAGE) of the crude venom of (**A**) *M. mipartitus*, (**B**) *M. dumerilii*, (**C**) *M. spixii*, (**D**) *M. surinamensis*, (**E**) *M. lemniscatus*, (**F**) *M. medemi*, (**G**) *M. dissoleucus*, (**H**) *M. isozonus*, and (**I**) *M. ancoralis* using a C18 column (250 mm–4.6 mm), an elution gradient used: 0–70% of acetonitrile (99% in TFA 0.1%). The run was monitored at 215 nm, and the assignation of the regions was made using (**A**) Rey-Suárez et al., 2011 [5], (**B**) Rey-Suárez et al. (2016) [6], and (**C**–**E**) Sanz et al., 2019 [20,21].

**Figure 2 toxins-16-00183-f002:**
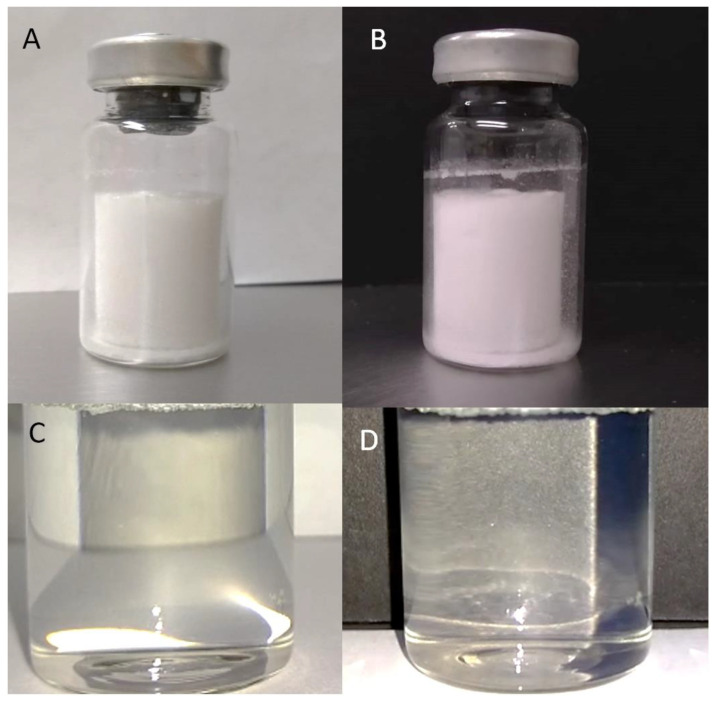
Appearance test of the antivenom product freeze-dried on a white background (**A**) and black background (**B**) and resuspended on a white background (**C**) and black background (**D**).

**Figure 3 toxins-16-00183-f003:**
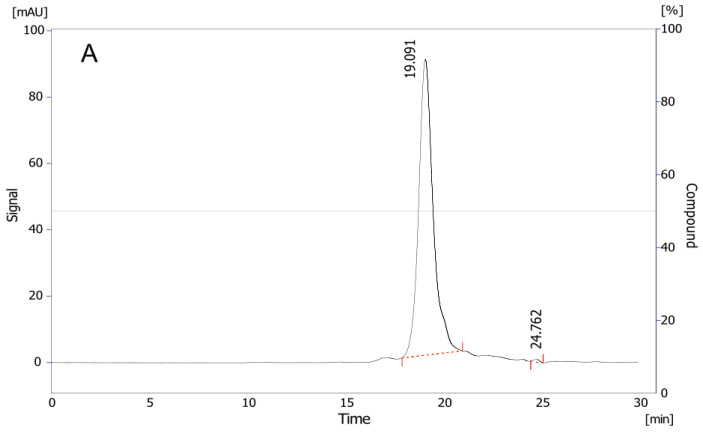
(**A**). SEC-HPLC of one freeze-dried polyvalent antivenom vial using a column Sec 3000 Phenomenex, mobile phase buffer Na_2_HPO_4_ 15 mM NaH_2_PO_4_ 30 mM NaCl 200 mM (pH 7.0), flux rate 0.5 mL/min, and detection of UV 280 nm. The red dashed line indicates the beginning and end peak elution time to measure the area under the curve (**B**). Electrophoresis of the freeze-dried polyvalent antivenom (Lines 1 and 2) and INS antivenom (Line 3) under non-reducing conditions using 12% sodium dodecyl sulfate–polyacrylamide gels (SDS-PAGE). Lowercase letters a–e indicate the recognized bands with the respective calculated MW using Gel Analyzer 19.1 [23].

**Figure 4 toxins-16-00183-f004:**
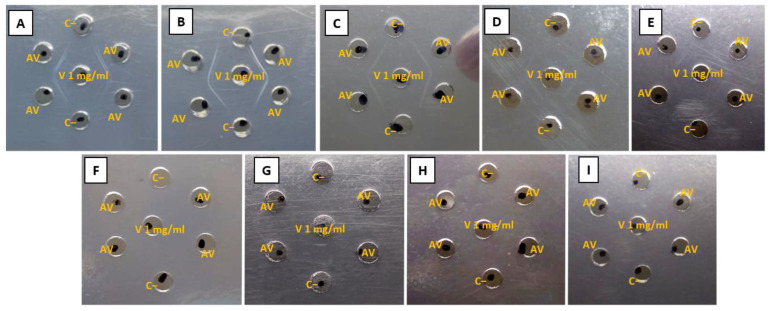
Double immunodiffusion test in agarose gel at 1% of (**A**) *M. mipartitus*, (**B**) *M. dumerilii*, (**C**) *M. isozonus*, (**D**) *M. dissoleucus*, (**E**) *M. ancoralis*, (**F**) *M. lemniscatus*, (**G**) *M. spixii*, (**H**) *M. medemi*, and (**I**) *M. surinamensis* venoms (V), 30 µL at a concentration of 1 mg/mL against antivenom of Universidad de Antioquia (AV), 30 µL at a concentration of 30 mg/mL, and water for injection as negative control (C−).

**Figure 5 toxins-16-00183-f005:**
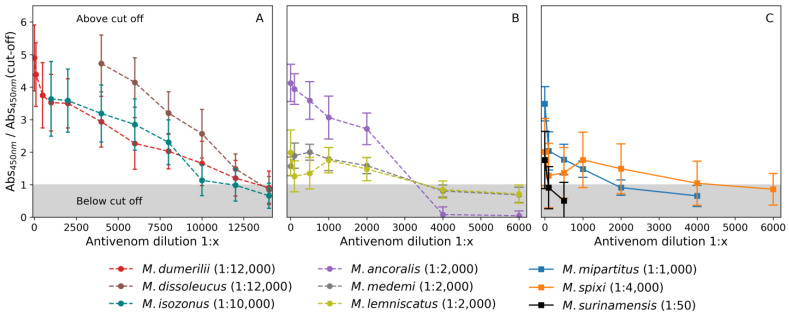
Antivenom recognition in ELISA test for the complete *Micrurus* venoms. The lines with circular (●) marks refer to species with PLA_2_s-predominant venoms and (**A**,**B**) squares (■) 3FTxs-predominant (**C**). The cut-off point was three times the capacity of recognition of the plasm of a non-immunized horse. Each dilution was tested in quadruplicate using two separate vials (8 repetitions in total).

**Table 1 toxins-16-00183-t001:** pH, dilution time, and extractable volume of the antivenom.

Vial	pH	Dilution 100% (Minutes:Seconds)	Extractable Volume (mL)
1	6.01	3:53	10.63
2	6.05	4:24	10.53
3	5.96	4:31	10.52
4	6.08	4:26	10.16
5	6.11	4:45	10.09
6	6.06	4:32	10.42
Average ± standard deviation	6.04 ± 0.05	4:25 ± 0:17	10.39 ± 0.19

**Table 2 toxins-16-00183-t002:** Total protein and albumin concentration of the antivenom product.

Vial	Total Protein mg/mL	Albumin mg/mL (%albumin/Protein)
1	33.82	0.377 (1.115)
2	33.57	0.377 (1.123)
3	31.37	0.333 (1.062)
4	28.74	0.421 (1.146)
5	29.11	0.377 (1.295)
6	29.18	0.377 (1.295)
Average ± standard deviation	30.97 ± 2.3	0.377 ± 0.02 (1.22% ± 0.15)
INS antivenom control	42.68	1.05 (2.46%)

**Table 3 toxins-16-00183-t003:** Efficacy of the antivenom product against the venom of *M. mipartitus* and *M. dumerilii*.

Venom Used (3xLD_50_)	Venom to Be Neutralized for 1 mL of Antivenom (µg)	Mice Survivor
*M. mipartitus*	100	3/3
	200	2/3
	300	1/3
	400	0/3
	232 (Confirmation test)	2/4
*M. dumerilii*	250	3/3
	350	2/3
	450	1/3
	600	3/3
	399 (Confirmation test)	2/4

**Table 4 toxins-16-00183-t004:** Comparative weight of some organs in mice submitted for the acute toxicity test.

Doses of Antivenom (mg)		Mice 1 (g)	Mice 1 (g)	Mice 1 (g)	Reference Value (g)	Value *p* in *t*-Student
9	Heart	0.130	0.154	0.120	0.155	0.198
	Spleen	0.055	0.099	0.067	0.118	0.082
	Kidneys	0.294	0.403	0.518	0.413	0.998
	Liver	1.320	1.280	1.450	1.486	0.114
12	Heart	0.124	0.124	0.159	0.155	0.240
	Spleen	0.079	0.112	0.097	0.118	0.149
	Kidneys	0.395	0.402	0.427	0.413	0.610
	Liver	1.340	1.384	1.204	1.486	0.086
15	Heart	0.153	0.138	0.166	0.155	0.785
	Spleen	0.052	0.107	0.074	0.118	0.129
	Kidneys	0.270	0.327	0.421	0.413	0.239
	Liver	1.020	1.385	1.258	1.486	0.133

## Data Availability

The data presented in this study are available in the article.

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
