# Peer review of "Standard Quality Characteristics and Efficacy of a New Third-Generation Antivenom Developed in Colombia Covering Micrurus spp. Venoms"

_toxins, 2024, doi:10.3390/toxins16040183_

Round 1

Reviewer 1 Report

Comments and Suggestions for Authors

 The manuscript “Standard quality characteristics and efficacy of a new third- generation antivenom developed in Colombia covering Micrurus spp. venoms” represents a significant contribution to the field of snakebite antivenom treatment that is considered a serious public health issue, lacking survey around the world. I am sure about the importance of this kind of contribution, once snakebite envenomation is considered a Neglected Tropical Disease by the World Health Organization Overall, the work is appropriately designed, well conducted, reported, and discussed. In addition, it can contribute to the venom and antivenom field. However, there are some concerns that deserve the authors attention.

Main revision:

1)    Considering that the study is based on the analysis of third generation Micrurus antivenom, in my opinion, it is essential to inform which species of snakes were used for the antivenom production. This information will help to better understand about the snake species venoms neutralization. For example, in lines 273-280, the knowledge of the snake species venoms used for horse immunization can explain better the reason of the recognizing results, besides the PLA2 and FTX content.

Minor revision:

-        I suggest to include one statement about the third antivenom generation in the introduction.

-        Line 38 – Remove an extra [.

-        Figure 4 – Snake species names must be in italic.

-        Line 37- “Letality” instead “lethality”.

So, I recommend accept after minor revision.

Author Response

We would like to thank the reviewers for their careful reading of the manuscript and the detailed comments, which have helped us improve it. In order to facilitate the revision and the verification of the comments inclusion in the manuscript we have listed each comment by separate and responded accordingly the requested information.

Comments.

Main revision:

  • Considering that the study is based on the analysis of third generation Micrurus antivenom, in my opinion, it is essential to inform which species of snakes were used for the antivenom production. This information will help to better understand about the snake species venoms neutralization. For example, in lines 273-280, the knowledge of the snake species venoms used for horse immunization can explain better the reason of the recognizing results, besides the PLA2 and FTX content.

R/ The aim of this research is to evaluate the standard quality characteristics and efficacy of a new third-generation antivenom developed in Colombia. This antivenom was supplied by the manufacturing team and the information about immunization was not supplied in order to avoid bias. As far as we know is that the immunogenic mixture complies neurotoxic venom including mipartoxin and PLA2 rich venoms.

 Minor revision:

  -        I suggest to include one statement about the third antivenom generation in the introduction.

R/ Corrected as suggested. Please check lines 77 to 85.

-        Line 38 – Remove an extra [.

R/ Corrected as suggested. Please check line 49.

-        Figure 4 – Snake species names must be in italic.

R/ Corrected as suggested. Please check lines 167 to 171.

-        Line 37- “Letality” instead “lethality”.

R/ Corrected as suggested. Please check line 112.

Reviewer 2 Report

Comments and Suggestions for Authors

The manuscript “ Standard quality characteristics and efficacy of a new third-generation antivenom developed in Colombia covering Micrurus spp. Venoms” proposed a work focused on the evaluation of the physiological and a series of preclinical parameters (following the WHO recommendations) such as appearance, solubility, extractable volume, pH, albumin and protein concentrations, identity, purity and integrity of immunoglobulins fragment, abnormal toxicity, neutralization efficacy against the venom of two Micrurus species and a cross-recognition against a third species. The authors have listed series of results regarding all the evaluated parameters listed above. This work follow a recent work published by Ana Cardona-Ruda and colleagues in Toxins in 2022 on the same field of research and on the same species that is not mentioned in the introduction of the proposed manuscript.

The authors performed an interesting peace of work with a lot of complementary methods to investigate this third-generation antivenom developed in Colombian to face the Micrurus snakebites.

Comments:
The reference list needs to be revised for homogeneity.

The Keyword needs to be improved in order to facilitate the identification of this manuscript if published. In general, It is better to avoid in the keyword list, words that are already in the manuscript title. Why not proposing Coral snake for example rather than Micrurus or even efficacy or freeze-dried.

The abstract needs to be revised to propose more pertinent information. The reviewer is not convinced by the sentence from line 8 to 101 as a primary information for the reader at least at the level of an Abstract. Try to avoid acronyms in the abstract., is it really reasonable to propose a 100% solubility in 4 minutes and 25 seconds (line 20). Line 23 Antivenom should be antivenom. There is no clear sentence in this abstract to conclude on the results they obtained, it is only a list of objectives and results listed without a clear message.

The Key contribution proposed is difficult to follow. The authors are mentioning line 28 “antimicrobial” blabla. Why is this record interesting as a third generation of an antivenom for Colombia?

The introduction needs to be revised to be clearer. Line 43: What did the authors mean by “showing a dichotomy across America”? Is it an important issue for their work’ Not certain this is essential? A minor cosmetic point line 41 [[3] should write [3]. The authors are mentioning that they are dealing with a third generation antivenoms, why not reporting in this session Introduction a brief literature on the first and second generations to help the reader in understanding what is really innovative in this third generation?

Materials and Methods: Section 5.1. What is the sex of the mice used for the in vivo experiments. DO the authors follow the animal wellbeing requirements? If yes, please specify this. Line 343 the authors are writing “non-GMP facility but under controlled process”. What are these conditions? What is the quantity of product used for the horse immunisation? It is not clear what is the source of the material used for immunising the horses. The venom obtention section is not detailed enough. What is the quantity of venom collected per snake? As the venom may vary geographically for example (also according to the milking frequency or after a certain duration post nutrition of the snakes) even within a same species how the authors are considering such a variability in the venom composition? Line 369, what was the criteria used to perform the hand collection of the fractions? Do the authors used the mAU variations (not according to the elution times)? Apparently, they collected only most of the major peaks and nothing between them. A clarification is needed. Lines, 373-374, change for “with markers covering the mass range from 250 kDa down to 10 kDa” instead of providing a long list of molecular weights. Same comment lines 462-463. What was the staining method used for the SDS-PAGE revelation? Line 383 5.4.2 Lethality and not lethality. Line 385 what is the positive control? Line 386 “LD50 in the literature” the reference should be listed. Section 5.5 is not clear. 5.5.5. and 5.5.6. How many replicates were done per vial? Line 472 size and not Size. The definition of SEC is size exclusion chromatography and not size exclusion. Revised this.

Result section. Line 82 acetonitrile and not Acetonitrile. Regarding the venom analysis (2.1.1: the authors are mentioning that they were able to collect different numbers of fraction according to the snake species (Figure 1). It is not clear what is a fraction, is it a peak or series of peaks eluted between a window of different retention times. The wavelength (215 nm) used to follow the elution pattern is different from the one used to follow the SEC elution pattern (280nm), why this? If it is peaks, then there are plenty of molecules that were not collected and analysed. Why this? Regarding the SDS-PAGE analyses, many fractions have molecules with molecular weights below the limit of separation. Why not performing molecular mass fingerprints (MFPs) by a method frequently used for such MFPs, the versatile MALDI mass spectrometry for a better understanding of the venom profils? Section 2.1.2: It is quite difficult why the authors are referring to LD50 values published in 1992, and not a more recent evaluation if available to compare with the new venom preparations. Are the values presented coming from replicates or a single evaluation? Results proposed in figure 2 are really not convincing (poor quality of the pictures). What is the sensitivity of the equipment used to monitor the pH? What was the temperature of the solution at the time of pH measurement? Was the dilution times measured several times per vial or just once per vial? Page 6, Table 2 should be revised to avoid non English writing. What is the INS control antivenom compared to the 6 antivenom vials? Different batch, duration of conservation, properties of this venom compared to the ones of the 6 vials? Apparently the control INS antivenom is rather different from the 6 ones used in this study (see Figure 3 line 3 of the SDS PAGE. How was defined the purity index? On figure 3A, it is clear that the base line consideration for the main peak eluted at 19.08 minutes is not properly selected. There is a clear shoulder at the end of the peak. Having the values of the surfaces of the two considered peaks would have been more informative. Why selecting the peak at the elution time of 24.75 minutes and the ones before and after the main peak that are clearly visible? Having a mass spectrometry record would have been much more confident. Figure 4, revised the legend to have the Latin names in italic.

The discussion session is very long and needs to be revised to deliver a clear conclusion. However, a s many results proposed are not totally convincing, their discussion needs to be more nuanced.

Conclusions. Line 315 9 or nine but not both. Line 143 page 7 section 2.3.3. The antivenom used for the Ouchterlony recognition is coming from which vial? 

Revised the entire manuscript for typo alterations and homogeneity (e.g. ml or mL) and English improvement.

For all these reasons, the manuscript as proposed is not acceptable for publication.

Author Response

We would like to thank the reviewers for their careful reading of the manuscript and the detailed comments, which have helped us improve it. In order to facilitate the revision and the verification of the comment's inclusion in the manuscript, we have listed each comment separately and responded accordingly to the requested information.

Comments.

  • This work follow a recent work published by Ana Cardona-Ruda and colleagues in Toxins in 2022 on the same field of research and on the same species that is not mentioned in the introduction of the proposed manuscript.

R/ As recommended, we added the Ana Cardona-Ruda and colleagues in Toxins in 2022 reference in the introduction in order to include information about the progress to improve antivenom production in the country. Please check line 85.

It is very important to consider that the objective of this manuscript is to publish the standard quality characteristic and efficacy results of a new antivenom developed in Colombia; Ana Cardona-Ruda and colleagues’ article is not directly related to our publication, our antivenom product is very different, and the only common species in both researches is M. mipartitus.

Comments:

  • The reference list needs to be revised for homogeneity.

R/ Reference list revised as recommended. All references were organized using a free reference manager and the citation style of the journal. Please check lines 553 and 658.

  • The Keyword needs to be improved in order to facilitate the identification of this manuscript if published. In general, It is better to avoid in the keyword list, words that are already in the manuscript title. Why not proposing Coral snake for example rather than Micrurus or even efficacy or freeze-dried.

R/ The keywords were improved as recommended. Please check line 34.

  • The abstract needs to be revised to propose more pertinent information.

R/ We improved the abstract section in order to show clear information about the research. It is very important to consider that the abstract was organized following the journal guidelines. Please check lines 6 to 32.

  • The reviewer is not convinced by the sentence from line 8 to 101 as a primary information for the reader at least at the level of an Abstract.

R/ We improved the abstract section to give a clear idea about the research aim in order to show the product parameters to be tested, the respective results and a clear conclusion. It is very important to consider that the abstract was organized following the journal guidelines. Please check lines 6 to 33.

  • Try to avoid acronyms in the abstract.

R/ Improved as suggested. Please check lines 6 to 33.

  • is it really reasonable to propose a 100% solubility in 4 minutes and 25 seconds (line 20).

R/ As a quality parameter suggested by the WHO and the local regulators in Colombia, the parameter should be evaluated, and the result needs to fulfil the rank value (less than 10 minutes). In the pharmaceutical industry, if the value exceeds the limit, the antivenom exhibits a quality deviation, and the batch may not be approved for further commercialization. In our case, the antivenoms show a solubility time within 3:53 to 4:45 minutes, fulfilling the parameter. Previous reports indicate that some antivenoms manufactured in Colombia use to solubilize after 20 minutes, which is absolutely unacceptable to guarantee an appropriate opportune treatment.

  • Line 23 Antivenom should be antivenom.

R/ Improved as suggested. Please check line 25.

  • There is no clear sentence in this abstract to conclude on the results they obtained, it is only a list of objectives and results listed without a clear message.

R/ We improved the abstract section to give a clear idea about the research aim in order to show the product parameters to be tested, the respective results and a clear conclusion. It is very important to consider that the abstract was organized following the journal guidelines. Please check lines 6 to 33.

  • The Key contribution proposed is difficult to follow. The authors are mentioning line 28 “antimicrobial” blabla. Why is this record interesting as a third generation of an antivenom for Colombia?

R/ The word “antimicrobial” is a typo (an autocorrection from the editor program), the correct word is antimicruric. The typo is corrected.

Introduction

  • The introduction needs to be revised to be clearer.

R/ The introduction covers key aspects of Micrurus biology and snakebites, including the presence of corals snakes in Colombia, venom composition and variability, envenoming, and current antivenom therapy.

  • Line 43: What did the authors mean by “showing a dichotomy across America”? Is it an important issue for their work’ Not certain this is essential?

R/ The mention of the dichotomy across the continent is essential to depict the venom composition variation across America and how it is observed in the available Micrurus species in Colombia; in the following sentence, we focus on the dichotomy of the Micrurus snakes across Colombia. In the design of a polyvalent antivenom, it is absolutely essential to consider any toxin variation (especially the one involved in the toxic manifestation of the snakebite) in the venom of the medicinally important snake species since antivenoms should be formulated to cover this divergence in the genera. Polyspecific antivenoms (polyvalent) are normally formulated when you want to cover different species (or genera) since is difficult to recognize or differentiate snakebites by clinical manifestations. In this case, M. mipartitus and M. dumerilli show different venom compositions, with M dumerilli lacking the expression of neurotoxic mipartoxin and exhibiting instead a rich content of PLA2. Despite the venom composition difference, the clinical manifestations of both snakebite are highly similar and difficult to differentiate.

  • A minor cosmetic point line 41 [[3] should write [3].

R/ Corrected as suggested. Please check line 76.

  • The authors are mentioning that they are dealing with a third generation antivenoms, why not reporting in this session Introduction a brief literature on the first and second generations to help the reader in understanding what is really innovative in this third generation?

R/ Information added as suggested. Please check lines 77 to 83.

Materials and Methods:

  • Section 5.1. What is the sex of the mice used for the in vivo experiments.

R/ Information was added in the section as suggested (males and females). Please check line 352.

  • DO the authors follow the animal wellbeing requirements? If yes, please specify this.

R/ For the animal wellbeing requirements, in lines 355 to 358, we mention the record number of the ethics committee for animal experimentation of the University of Antioquia approved for the paper and the use of the 3Rs protocols.

  • Line 343 the authors are writing “non-GMP facility but under controlled process”. What are these conditions?

R/ The “controlled process” is not an infrastructure condition, it is an activity related to antivenom production. In any pharmaceutical process, you must control it in terms of the most important variables. We are not talking about standardization or validation; it is only the control of the main variable, which we have identified as the critical process parameters (CPP) affecting the critical quality attribute (CQA) of the antivenom in terms of quality target product profile (QTPP).

  • What is the quantity of product used for the horse immunisation? It is not clear what is the source of the material used for immunising the horses.

R/ This research aims to evaluate the standard quality characteristics and efficacy of a new third-generation antivenom developed in Colombia. This antivenom was supplied by the manufacturing team, and the immunization information was not supplied in order to avoid bias. As far as we know, the immunogenic mixture complies with neurotoxic venom, including mipartoxin and PLA2-rich venoms.

  • The venom obtention section is not detailed enough. What is the quantity of venom collected per snake? As the venom may vary geographically for example (also according to the milking frequency or after a certain duration post nutrition of the snakes) even within a same species how the authors are considering such a variability in the venom composition?

R/ The venoms used in the project were supplied by the serpentarium of the University of Antioquia and the venom bank of the Antivenom project from the same University. We described the way the venom is obtained and handled in the serpentarium and the quantities obtained after each manual extraction were not supplied neither requested since this data is not currently necessary to evaluate the standard quality characteristics and efficacy of a new third-generation antivenom developed in Colombia. Further, the data will be useful after an antivenomic study to analyze the specific neutralizing capacity of the antivenom by venom toxin. The geographical distribution is supplied. Please check line 375.

  • Line 369, what was the criteria used to perform the hand collection of the fractions? Do the authors used the mAU variations (not according to the elution times)? Apparently, they collected only most of the major peaks and nothing between them. A clarification is needed.

R/ Corrected as suggested. We added the specification of the collection of fractions over 100 mAU were collected. Please check line 390.

  • Lines, 373-374, change for “with markers covering the mass range from 250 kDa down to 10 kDa” instead of providing a long list of molecular weights. Same comment lines 462-463.

R/ Corrected as suggested. Please check lines 396 and 486.

  • What was the staining method used for the SDS-PAGE revelation?

R/ The FastGene Q-Stain was the methodology for the SDS-PAGE staining. Please check lines 392 and 486.

  • Line 383 5.4.2 Lethality and not lethality.

R/ Corrected as suggested. Please check line 403.

  • Line 385 what is the positive control?

R/ Correction was applied, and information was added (negative controls to which only 200 µL of saline solution was) in line 406.

  • Line 386 “LD50 in the literature” the reference should be listed.

R/ The literature LD50 is shown below in the same paragraph in lines 410 to 411.

  • Section 5.5 is not clear.

R/ Corrected. Please check lines 414 to 419.

  • 5.5. and 5.5.6. How many replicates were done per vial?

R/ Corrected as suggested. We used 3 replicates per assay. Please check lines 454 and 466.

  • Line 472 size and not Size.

R/ Corrected as suggested. Please check line 494.

  • The definition of SEC is size exclusion chromatography and not size exclusion. Revised this.

R/ Corrected as suggested. Please check line 494.

Result section.

  • Line 82 acetonitrile and not Acetonitrile.

R/ Corrected as suggested. Please check line 99.

  • Regarding the venom analysis (2.1.1: the authors are mentioning that they were able to collect different numbers of fraction according to the snake species (Figure 1). It is not clear what is a fraction, is it a peak or series of peaks eluted between a window of different retention times.

R/ In Chromatography, venoms (and other complex protein mixtures) are fractionated in different fractions according to the differential partition or interaction between two phases (a mobile phase and a stationary phase). A fraction is a portion of something observed as a whole; in this case, the whole venom is fractionated by chromatography in smaller fractions which are collected. After detection, the systems integrates the detected signals and show it as peaks in a chromatogram. Peaks are signals observed in the chromatogram.

  • The wavelength (215 nm) used to follow the elution pattern is different from the one used to follow the SEC elution pattern (280nm), why this?

R/ Both wavelengths can be used when you are analysing venoms. Wavelength analysis at 214-215 nm detects peptidic bonds, while 280 nm allows the detection of proteins with aromatic residues. When the sample is a complex mixture of different proteins and/or peptides and you know the content, it is better to analyse the chromatographic run using a wavelength that allows you to detect all protein kinds of compounds (215 nm), even those without aromatic residues. In the case of the SEC analysis, we analysed a known molecule, the F(ab)2 fragment, which has aromatic residues and that is we used 280 nm in this case.

  • If it is peaks, then there are plenty of molecules that were not collected and analysed. Why this? Regarding the SDS-PAGE analyses, many fractions have molecules with molecular weights below the limit of separation. Why not performing molecular mass fingerprints (MFPs) by a method frequently used for such MFPs, the versatile MALDI mass spectrometry for a better understanding of the venom profils?

R/ We did not perform any molecular mass analysis because we currently do not have access to mass spectrometry equipment, and we cannot afford the service with an external laboratory.

  • Section 2.1.2: It is quite difficult why the authors are referring to LD50 values published in 1992, and not a more recent evaluation if available to compare with the new venom preparations.

R/ It is very important to reduce the number of animals used in research activities. Additionally, the Micrurus venom is a highly valuable and limited resource. What we did is to confirm the previously published LD50. The LD50 of the animal was the same as currently used in this test; the doses were checked in section 2.1.2.

  • Are the values presented coming from replicates or a single evaluation?

R/ The LD50 confirmation analysis was performed one single time to reduce the animal's use.

  • Results proposed in figure 2 are really not convincing (poor quality of the pictures).

R/ We improve the resolution of the pictures.

  • What is the sensitivity of the equipment used to monitor the pH? What was the temperature of the solution at the time of pH measurement?

R/ The sensitivity is 0,01, and the temperature is 26 degrees Celsius. Please check line 442.

  • Was the dilution times measured several times per vial or just once per vial?

R/ It is impossible to measure the dilution times more than once without realising a second lyophilised process.

  • Page 6, Table 2 should be revised to avoid non English writing.

R/ Corrected as suggested. Please check line 135.

  • What is the INS control antivenom compared to the 6 antivenom vials? Different batch, duration of conservation, properties of this venom compared to the ones of the 6 vials? Apparently the control INS antivenom is rather different from the 6 ones used in this study (see Figure 3 line 3 of the SDS PAGE.

R/ Corrected as suggested. Please check lines 78 and 135. Clarification was made in the introduction.

  • How was defined the purity index? On figure 3A, it is clear that the base line consideration for the main peak eluted at 19.08 minutes is not properly selected. There is a clear shoulder at the end of the peak. Having the values of the surfaces of the two considered peaks would have been more informative. Why selecting the peak at the elution time of 24.75 minutes and the ones before and after the main peak that are clearly visible? Having a mass spectrometry record would have been much more confident.

R/ The purity was estimated based on two results, not only the SEC. The SEC gave us a very close idea about the index of purity, but we were only able to see two peaks in the chromatogram, while the electrophoresis showed us at least 3 different bands. We also considered the albumin concentration, which is the lower value of purity regarding the protein concentration. Our analysis is based on the techniques requested by the WHO and the Colombian regulator.

  • Figure 4, revised the legend to have the Latin names in italic.

R/ Corrected as suggested. Please check lines 167 to 171.

Discussion

  • The discussion session is very long and needs to be revised to deliver a clear conclusion. However, a s many results proposed are not totally convincing, their discussion needs to be more nuanced.

R/ All the sections in the manuscript were improved as suggested.

Conclusions.

  • Line 315 9 or nine but not both.

R/ Corrected as suggested. Please check lines 333 to 349.

  • Line 143 page 7 section 2.3.3. The antivenom used for the Ouchterlony recognition is coming from which vial? 

R/ We used a new vial.

  • Revised the entire manuscript for typo alterations and homogeneity (e.g. ml or mL) and English improvement.

R/ Corrected as suggested.

Round 2

Reviewer 2 Report

Comments and Suggestions for Authors

The authors are proposing a revised version according improved but for the reviewer not of the quality expected. For many moments the authors are proposing to the reviewer a quid of tutorial on what is a fraction what is an appropriate wavelength to record chromatograms. Apparently they did not understand the question. The reviewer was not asking the definition of a fraction in chromatography but how fractionation was performed by time or by absorbency and when the authors were decided the beginning of a fraction and the end. They are different ways of fractioning a sample by HPLC, etc. There is still a lack of statistical robustness.

Author Response

We would like to thank the reviewers for their second reading of the manuscript and the detailed comments, which have helped us to continue to improve it. In order to facilitate the revision and the verification of the comment's inclusion in the manuscript, we have listed each comment separately and responded accordingly to the requested information.

  • The authors are proposing a revised version according improved but for the reviewer not of the quality expected. For many moments the authors are proposing to the reviewer a quid of tutorial on what is a fraction what is an appropriate wavelength to record chromatograms. Apparently they did not understand the question. The reviewer was not asking the definition of a fraction in chromatography but how fractionation was performed by time or by absorbency and when the authors were decided the beginning of a fraction and the end. They are different ways of fractioning a sample by HPLC, etc.

R/ The collection was realized manually by looking at the absorbance; the correction was added in lines 391 to 393. We follow the description of the process realized by Rey-Suárez et al., 2016.

  • There is still a lack of statistical robustness.

R/ We made some clarification of the statistical analysis in lines 456-457, 468-469, 521-525, and 534 and added the statistical analysis methodology in lines 549 to 553 “Physicochemical results are expressed as mean ± standard error media (S.E.M.) after a t-student analysis using Excel and the programming learn Python, with the library Matplotlib (graphics), while in-vivo preclinical tests were analyzed using the library Statsmodels (probit analysis)

Bibliography

Rey-Suárez, P., Núñez, V., Fernández, J., & Lomonte, B. (2016). Integrative characterization of the venom of the coral snake Micrurus dumerilii (Elapidae) from Colombia: Proteome, toxicity, and cross-neutralization by antivenom. Journal of Proteomics, 136, 262–273. https://doi.org/10.1016/j.jprot.2016.02.006
